# Codon Usage Bias Analysis in Macronuclear Genomes of Ciliated Protozoa

**DOI:** 10.3390/microorganisms11071833

**Published:** 2023-07-18

**Authors:** Yu Fu, Fasheng Liang, Congjun Li, Alan Warren, Mann Kyoon Shin, Lifang Li

**Affiliations:** 1Laboratory of Marine Protozoan Biodiversity and Evolution, Marine College, Shandong University, Weihai 264209, China; fy_sdu@163.com (Y.F.); lfsdante@163.com (F.L.); licongjun123june@163.com (C.L.); 2Department of Life Sciences, Natural History Museum, London SW7 5BD, UK; a.warren@nhm.ac.uk; 3Department of Biology, University of Ulsan, Ulsan 44610, Republic of Korea; mkshin@ulsan.ac.kr

**Keywords:** codon usage bias, ciliates, macronuclear genome, mutation pressure, natural selection

## Abstract

Ciliated protozoa (ciliates) are unicellular eukaryotes, several of which are important model organisms for molecular biology research. Analyses of codon usage bias (CUB) of the macronuclear (MAC) genome of ciliates can promote a better understanding of the genetic mode and evolutionary history of these organisms and help optimize codons to improve gene editing efficiency in model ciliates. In this study, the following indices were calculated: the guanine-cytosine (GC) content, the frequency of the nucleotides at the third position of codons (T3, C3, A3, G3), the effective number of codons (ENc), GC content at the 3rd position of synonymous codons (GC3s), and the relative synonymous codon usage (RSCU). Parity rule 2 plot analysis, Neutrality plot analysis, ENc plot analysis, and correlation analysis were employed to explore the main influencing factors of CUB. The results showed that the GC content in the MAC genomes of each of 21 ciliate species, the genomes of which were relatively complete, was lower than 50%, and the base compositions of GC and GC3s were markedly distinct. Synonymous codon analysis revealed that the codons in most of the 21 ciliates ended with A or T and four codons were the general putative optimal codons. Collectively, our results indicated that most of the ciliates investigated preferred using the codons with anof AT-ending and that codon usage bias was affected by gene mutation and natural selection.

## 1. Introduction

Codons are nucleotide triplets of messenger RNA that carry genetic information. In all organisms, there are 64 kinds of codons, including three kinds of stop codons and 61 kinds of codons encoding 20 amino acids. The phenomenon that the same amino acid is encoded by more than one synonymous codon is called the degeneracy of the codon [1]. Except for methionine (Met) and tryptophan (Trp) which are encoded by ATG and TGG respectively, the other amino acids are encoded by 2–6 synonymous codons [2]. The non-randomness of synonymous codons is called codon usage bias (CUB). CUB can be mainly caused by base preference and natural selection in the genome, which is the result of the balance of mutation, natural selection, and genetic drift [3,4,5,6].

CUB differs not only among different organisms but also within the genome of the same species, even in a single gene. Many studies have shown that CUB is associated with a number of biological factors, including tRNA content [7,8,9], gene length [10,11], gene expression level [12,13], biased gene conversion [14,15,16], recombination rate [17,18,19], gene translation initiation signals [20,21], patterns of amino acid usage [22,23], GC content [24,25,26], mRNA folded stability and secondary structure [27,28], and gene location [29]. CUB plays an important role in many cellular processes, such as genome transcription [30], selection for optimized translation [31], the efficiency and accuracy of translation [32,33], and the structure, expression and function of proteins [34]. Highly expressed genes have stronger codon bias [35,36], therefore the study of CUB is helpful for determining the optimal codon, constructing gene expression vectors, and improving gene expression efficiency and transcription levels [37,38]. Furthermore, CUB has a profound impact on the design of transgenesis, the symbiotic relationships between pathogens and hosts, and the exploration of biomolecular evolution [39,40,41].

Research on CUB is well established in many organisms [42,43]. Ciliates are the most specialized and complex group of protozoa. They are widely distributed in marine, freshwater, and terrestrial habitats worldwide and several species (e.g., *Vorticella microstoma* and *Litonous lamella*) are reliable indicators of environmental quality [44,45,46]. The great majority of ciliates possess two distinct types of nuclei, i.e., the macronucleus (MAC) and micronucleus (MIC), which differ both in morphology and function. Furthermore, several ciliates, e.g., *Tetrahymena thermophila* and *Paramcium tetraurelia,* serve as model organisms and play an important role in molecular biology research. Research on codon usage has been carried out in *T. thermophila*, *P. tetraurelia*, and *P. biaurelia* [47,48,49]. In addition, differences in the nucleotide composition and CUB of the mitochondrial genome sequence in *P. tetraurelia* and *P. caudatum* have been reported despite the close relationship between these two species [50]. CUB of specific genes, including the highly expressed genes in *T. thermophila* and two centrin genes from *Entodinium caudatum,* have also been reported [51,52]. However, the general codon usage pattern of ciliates has not been fully investigated. Ciliates show different gene expression levels, physiological changes, and stress responses in different environments. Improved knowledge of CUB in ciliates will help to better understand their molecular mechanisms of adaptation to environmental change. The main aims of this study are to investigate codon usage patterns and the influencing factors of CUB in 21 species of ciliates representing four classes and two subphyla.

## 2. Materials and Methods

### 2.1. Data Sources

We analyzed codon usage bias of the coding sequences (CDSs) in 21 ciliates belonging to four classes of Ciliophora, namely Heterotrichea, Spirotrichea, Litostomatea, and Oligohymenophorea. The datasets were downloaded from the NCBI GenBank database (from https://www.ncbi.nlm.nih.gov/, accessed on 12 May 2022) (Appendix A). The CDSs were identified to remove unknown bases, repeated sequences, stop codons in the middle of sequences, and the codons of Met (methionine) and Trp (tryptophan) by Perl scripts [53]. Finally, each CDS should be an exact multiple of three bases and longer than 300 bp with complete start codons (ATG) and stop codons (TAA, TAG or TGA). For examples of different stop codons in ciliates, see [54].

### 2.2. Nucleotide Composition Analysis

The codon nucleotide composition index was calculated using BCAWT (Bio Codon Analysis Workflow Tool) [55], including the genome-wide GC content, the GC content at the 1st, 2nd and 3rd codon positions (GC1, GC2, GC3), the mean GC content at the 1st and 2nd codon positions (GC12), and the nucleotide composition in the 3rd codon position (A3, T3, C3, G3).

### 2.3. Effective Number of Codons

The effective number of codons (ENc) is a typical parameter to measure the magnitude of synonymous codon usage bias for any gene [55]. The ENc value quantifies synonymous codons usage frequency of a gene and is independent of gene length or amino acid composition [56]. ENc values range from 20 (extreme bias for using one codon) to 61 (no bias for using synonymous codons). In general, an ENc value < 35 is considered a significant CUB for the gene in question [56,57].

The Enc was calculated using the formulation of codon family (*F*_CF_) in the equation given by [58]:$$F_{CF}=\sum_{i=1}^{m}\;\;\left(\frac{n_{i}+1}{n+m}\;\right)2$$

Then, the ENc could be calculated by:$${ENc}_{.CF}=\frac{1}{F_{CF}}$$
where *n_i_* is the count of codon *i* in *m* amino acid family and *m* is the number of codons in an amino acid family. The subscript CF stands for “codon family” and refers to the fact that *F*_CF_ and ENc._CF_ are for a specific codon family rather than for a gene.

ENc values were plotted against GC3s values to reveal the determinants of CUB, thereby indicating whether there are other factors affecting CUB. The standard curve of the plot was calculated by the following formula [58]:$$ENc=2+GC3s+\frac{29}{GC3s^{2}+{\left(1-GC3s\right)}^{2}}$$

If codon usage is limited only by GC mutation bias, the predicted ENc value will be on or near the standard curve. If the predicted ENc values are considerably far from the standard curve, the CUB is mainly affected by natural selection.

The ENc ratio was calculated according to the following formula:$${ENC}_{ratio}=\frac{\left({ENC}_{exp}-{ENC}_{obs}\right)}{{ENC}_{exp}}$$

The ENc_ratio_ value shows the difference between observed and expected ENc values [59,60].

### 2.4. Codon Adaptation Index (CAI)

The CAI is an effective measure for the relative adaptiveness of CUB in one gene compared with highly expressed genes [61]. A high CAI value indicates a stronger CUB and a higher expression level. The CAI value ranges from 0 to 1 according to the gene expression level. CAI was calculated by the equation [61]:$$CAI=exp\frac{1}{L}\sum_{k=1}^{L}\ln\omega_{c\left(k\right)}$$
where *L* is the count of codons in the gene and $\omega_{c\left(k\right)}$ is the ω relative adaptiveness value for the *k*-th codon in the gene [61]. The CAI value was calculated using BCAWT.

### 2.5. Relative Synonymous Codon Usage (RSCU) and Putative Optimal Codons

The RSCU value for the codon was analyzed as the ratio of the observed frequency of a codon to the expected frequency under the assumption that all synonymous codons of a particular amino acid are used equally, and the RSCU value is unaffected by gene length and amino acid frequency [62]. RSCU directly reflects the CUB. If the RSCU value of a codon is lower than 1, this means that the codon in question is used less frequently than average; if the RSCU value equals 1, this indicates that codon usage is unbiased; if the RSCU value is higher than 1, this means that the codon in question is used more frequently than average. Similarly, codons with RSCU values higher than 1.6 and lower than 0.6 are considered to be over-represented and under-represented in the CDS, respectively [63]. The equation to calculate the RSCU is [61]:$$RSCU=\frac{O_{ac}}{\frac{1}{{\;k}_{a}\;}\sum_{c\in C_{a}}O_{\begin{array}{c} ac \end{array}}}$$
where *O_ac_* is the count of codon *c* for an amino acid and *k_a_* is the number of synonymous codons. The RSCU value was calculated using BCAWT.

The putative optimal codons were determined by BCAWT. If each synonymous codon of an amino acid family is correlated with the Enc of all genes, we defined the optimal codon of each amino acid family as the codon with the strongest negative correlation between the RSCU and ENc values [55].

### 2.6. Grand Average Hydropathicity (Gravy) and Aromaticity (Aroma) Indices

Changes in Gravy and Aroma reflect variations in the number of amino acids used. The Gravy was calculated as the arithmetic average of the sum of the hydrophilic indices of each amino acid, with scores ranging from −2 to 2, i.e., positive values for hydrophobic proteins and negative values for hydrophilic proteins [64]. Aroma refers to the aromatic properties of proteins. It represents the frequency of the aromatic amino acids (phenylalanine, tyrosine, and tryptophan) in the translated gene product [65]. Gravy and Aroma indices were calculated using BCAWT.

### 2.7. Correspondence Analysis

Correspondence analysis is a multivariate statistical method that we applied to 59 codons (the exceptions being ATG, TGG, TAA, TAG, and TGA) to investigate the major trends in codon usage variation in all CDSs. Correspondence analysis plotted the distribution of genes and codons on a continuum of 59 dimensions based on the trends that affect the usage of synonymous codons in the genome [55]. The first axis (axis 1) represents the majority of variation in codon usage, and subsequently, the amount of variation explained by each axis gradually decreases [66].

### 2.8. Parity Rule 2 (PR2) Plot Analysis

The base composition at the 3rd codon position has extensive heterogeneity in the genomes of higher eukaryotes. We can analyze whether the factors affecting CUB are only random mutations (A3 = T3, G3 = C3), or mutations with selection (A3 ≠ T3, G3 ≠ C3). A PR2-plot that used two-fold, four-fold, and six-fold degenerate codon families utilized A3/(A3 + T3) as the vertical axis and G3/(G3 + C3) as the horizontal axis. A = T and G = C were the central positions and the coordinates were (0.5, 0.5) [67].

### 2.9. Translational Selection Index

The translation selection (P2) index measures the degree of bias of anticodon-codon interactions and thus can indicate translation efficiency [68]. Similar to PR2 which reflects the selection of cytosine and thymidine as the 3rd base of the codon, P2 is based on the different usage of homologous tRNA species in the process of gene translation to indicate CUB in the gene [12]. P2 was calculated according to the following formula [68]:$$P2=\frac{WWC+SST}{WWY+SSY}$$
where W = A or T, S = C or G, Y = C or T.

### 2.10. Neutrality Plot Analysis

GC content at the 3rd codon position is almost neutral to natural selection, whereas GC content at the 1st and 2nd codon positions is negatively affected by directional mutational pressure and selective restriction, respectively [69]. Thus, neutrality plots were illustrated by GC3 on the horizontal axis and GC12 on the vertical axis. The slope of the regression line indicated a neutral degree of GC content. If the slope of the line is close to or equal to ±1, this indicates that mutation pressure is the sole determinant of CUB. In contrast, if the slope of the line is close to 0, this indicates that natural selection is the sole determinant of CUB. When the slope of the regression line is equal to ± 1/2, it means that mutation pressure and selective constraints are equal [70].

### 2.11. Statistical Analysis

Correlation analysis was done using IBM SPSS Statistics 26 software. The figures were constructed using BCAWT version 1.0.0, GraphPad Prism version 8.0, and R software version 3.6.3.

### 2.12. Phylogenetic Analysis

The orthologous proteins alignment of the 21 ciliates species generated by Orthofinder [71]. The maximum likelihood (ML) tree was generated using RAxML version 8.2.12 (-x 12,345 -p 12,345 -m PROTCATLGF -N 1000 -f a) [72]. The Bayesian inference (BI) analysis was performed using PhyloBayes-MPI version 1.4 (-cat -gtr -x 10 10,000) [73].

## 3. Results

### 3.1. Nucleotide Compositions

CUB may be shaped by nucleotide composition bias, specifically the GC content of CDSs. Genomic GC content determined by mutational processes was the prime factor of codon usage variation across species, and it was important evidence that mutation pressure determined CUB [74,75]. In this study, genomic GC content and GC content at different codon positions of the 21 ciliate species are shown in Appendix A and Table 1. The GC content varied greatly among the four classes of ciliates investigated. *Strombidium stylifer* in the class Spirotrichea had the highest GC content (49.74%). In addition, the GC content of *Pseudokeronopsis flava*, *Pseudokeronopsis carnea* and *Halteria grandinella* in the class Spirotrichea was each over 40% (46.55%, 45.18% and 44.34%, respectively). The GC content of species belonging to the classes Litostomatea, Oligohymenophorea and Heterotrichea were relatively low, ranging from 23.49% to 32.75%. Furthermore, the GC contents at different codon positions (GC1, GC2, GC3) of the 21 ciliate species ranged from 32.83% to 52.32%, 24.20% to 37.48%, and 12.8% to 61.98%, respectively. There was a difference in GC content at different codon positions of the whole CDSs among the 21 ciliate species. The largest difference was found in the GC content of the 3rd codon position. These data indicated that there were differences in genome nucleotide compositions among different ciliate species. The nucleotide compositions at the 3rd codon base (A3, T3, C3, G3) in the 21 ciliate species were also analyzed (Table 1 and Appendix A). The A3 content of *E. caudatum* (class Litostomatea) was the highest (42.95%), while that of *S. stylifer* (class Heterotrichea) was the lowest (19.40%). The T3 content of *Uronema marinum* (class Oligohymenophorea) was the highest (45.43%), while that of *S. stylifer* was the lowest (18.61%). *Stylonychia stylifer* had the highest C3 content (35.78%), while *E. caudatum* had the lowest (7.61%). *Stylonychia stylifer* also had the highest G3 content (26.20%), while *U. marinum* had the lowest (4.95%). These findings suggested that for the 21 species investigated, the composition at the 3rd base of codon varied greatly within species and among classes.

### 3.2. Effective Number of Codons and its Association with GC3

The effective number of codons (ENc) is used to measure the CUB in a gene, and the codon bias degree increases with the decline of the ENc value [56]. The ENc values in the 21 ciliate species ranged from 31.48 ± 2.55 to 44.86 ± 5.48 (Table 2, mean ± SD), with an average value of 38.34 (SD = 4.0120). A lower ENc value indicated that there was a strong CUB in the ciliates (average ENc value was approximately 35), but the different ciliate species codon usage patterns were also remarkably distinct, i.e., the maximum ENc value was 44.86, while the minimum ENc value was 31.48. The low ENc value of the 21 ciliate species codons revealed the instability and evolutionary diversity of the ciliate genome. An ENc-plot was constructed with the ENc value of each ciliate species on the x-axis and GC3s on the y-axis (Figure 1). According to GC3s content, the ciliates were divided into some with low GC3s content, ranging from 12.80% to 36.71%, and others with high GC3s content, ranging from 47.61% to 61.98% (Table 1). The average value of GC3s was 29.46% (SD: 13.6437). As shown in Figure 1, some genes were located near or on the standard curve, suggesting that their CUB was mainly affected by mutation pressure. However, most of the genes were located above or below the standard curve, suggesting that other factors, such as natural selection together with mutation pressure, determined CUB. In addition, a significant correlation between ENc and GC3s was observed (Appendix A), suggesting that mutation pressure had a significant influence on CUB in the ciliates.

The ENc ratio, as given by (ENc_exp_ − ENc_obs_)/ENc_exp_, was calculated to show the difference between ENc_obs_ and ENc_exp_ more clearly. The ENc ratio was in the range of 0.1 to 0.3 (Appendix A), indicating that the ENc_exp_ values of most genes were significantly different from the ENc_obs_ values. These data suggested that although the ciliate CUB was related to differences in GC3s, it was mainly affected by other factors such as natural selection.

### 3.3. Relative Synonymous Codon Usage (RSCU) and Putative Optimal Codons

The RSCU value can reveal the codon usage pattern of the gene. RSCU values <1, =1, and >1 indicate that the frequency of codon usage is below, equal to, or above average values, respectively. Codons with RSCU values >1.6 and <0.6 were considered over-represented and under-represented, respectively [63]. The RSCU values of 59 codons (the exceptions being ATG, TGG, and three stop codons) in the 21 ciliate species were analyzed to show if there were differences among the four classes represented.

The RSCU values of the classes Oligohymenophorea, Litostomatea, and Heterotrichea were similar as shown in Figure 2 and Appendix A. In the class Oligohymenophorea, there were 28 codons with RSCU value > 1 (A:12, T:15, G:1, C:0, ending in A, T, C, G, respectively), 27 codons of which ended in A/T, accounting for 96.43% of all codons. In the class Heterotrichea, there were 32 codons with RSCU value > 1 (A:13, T:15, G:4, C:0), 28 codons of which ended in A/T, accounting for 87.50% of all codons. In the class Litostomatea, there were 26 codons with RSCU value > 1 (A:12, T:14, G:0, C:0), 26 codons of which ended in A/T, accounting for 100% of all codons. In the class Spirotrichea, however, the species whose GC contents were more than 40% (*S. stylifer*, *Pseudokeronopsis flava*, *P. carnea*, and *H. grandinella*) had 48 codons with RSCU value > 1 (A:11, T:14, G:8, C:15), 23 codons of which ended in G/C, accounting for 47.92% of all codons; the other species, which had GC content ranging from 30% to 40% (*Euplotes vannus*, *Euplotes octocarinatus*, *Oxytricha trifallax*, and *Stylonychia lemnae*) had 29 codons with RSCU value > 1 (A:12, T:14, G:2, C:1), 26 codons of which ended in A/T, accounting for 89.66% of all codons. In 20 ciliate species (the exception being *P. flava* in the class Spirotrichea), the most preferred codon was AGA encoding arginine. The results showed that the ciliate species of the classes Oligohymenophorea, Litostomatea, and Heterotrichea, and some of those in the class Spirotrichea (*E. vannus*, *E. octocarinatus*, *O. trifallax*, and *S. lemnae*), preferred using codons ending in A/T, whereas the other species of the class Spirotrichea, including *S. stylifer*, *P. flava*, *P. carnea* and *H. grandinella*, preferred using codons ending in G/C, which is consistent with the bias for the 3rd base of codons in different ciliate classes [26]. The RSCU was affected by the restriction of nucleotide composition, suggesting that mutation pressure was one of the most impactful factors of CUB.

The method described in [58] was used to determine the putative optimal codons (Appendix A). In the class Oligohymenophorea, the putative optimal codons of 17 of the 18 amino acids ended in A/T, the exception being lysine (which was coded by AAG in *Tetrahymena borealis* and *T. elliotti*) and phenylalanine (which was coded by TTC). The putative optimal codons of the 18 amino acids in the classes Litostomatea and Heterotrichea all ended in A/T. By contrast, in the class Spirotrichea, there were 15 amino acids in *S. stylifer*, 18 in *P. flava*, 17 in *P. carnea*, and five in *H. grandinella*, whose putative optimal codons ended in G/C. There were differences in CUB among different classes of the 21 species investigated here, indicating that ciliates may be restricted by CUB in the evolution process.

### 3.4. PR2-Plot Analysis

PR2 is an intrastrand rule where A = T and G = C are expected if there is no mutation pressure or selection bias. If the usage of AT and CG are unbalanced, then both natural selection bias and mutation pressure together determine the composition of synonymous codons at the 3rd codon position and influence the CUB [76]. In most protein-coding genes, there are wide differences between both C and G content and A and T content [77]. We observed that the genes were distributed in four regions in the PR2-plot (Figure 3). In the 21 ciliate species, the AT bias ranged from 43.86% to 52.45% (Appendix A), and only five species had a higher AT bias than 50% (*P. caudatum*, *P. tetraurelia*, *E. octocarinatus*, *E. vannus*, and *S. stylifer*). The GC bias ranged from 36.89% to 58.35% and only *S. stylifer* had a GC bias higher than 50%. Thus, in the 21 species, the rate of codon usage ending in T/C was higher than that ending in A/G, which was consistent with the nucleotide composition in species where the 3rd codon position ending in pyrimidine bases was preferred. This finding was also supported by correlation analysis between ENc and A, T3, C3, G3, which showed a more significant correlation between ENc and T3, C3 (Appendix A). A codon usage imbalance between A/T and G/C as shown in the PR2-plot suggested that both natural selection bias and mutation pressure worked together on CUB in the 21 species.

### 3.5. Neutrality Plot Analysis

The difference in GC3 among the different species reflects the mutation pressure [78]. A neutrality plot analysis, which shows the relationship between GC12 and GC3, was conducted in the 21 ciliate species to explore the influence of mutation pressure and selection bias on CUB. There was a significant correlation between GC12 and GC3 (Appendix A), meaning that mutation pressure had a significant effect on CUB. Furthermore, the absolute value of the slope of the regression line in the neutrality plot ranged from 0.020 to 0.377, which indicated the effect of mutation pressure was only about 2% to 37.7% (Figure 4). The above results showed that although the CUB was affected by mutation pressure, natural selection seemed to have a greater influence. Four species (*E. caudatum*, *P. persalinus*, *Stentor roeselii,* and *S. stylifer*) with higher mutational pressure may have more rapid rates of evolution and higher adaptability than the other species investigated.

### 3.6. Correspondence Analysis

Correspondence analysis creates a series of orthogonal axes to determine the tendency to explain variation in data, with each subsequent axis explaining a gradual decrease in the amount of variation [79]. RSCU correspondence analysis in the 21 ciliate species was carried out show the trend of CUB based on RSCU values. In order to minimize the influence of amino acid composition on codon usage, each gene was represented as a vector with 59 dimensions, and each dimension corresponded to the RSCU value of a justice codon (excluding Met, Trp, and three stop codons) [80]. The first axis (axis 1) of the 21 species contributed 5.82% to 37.59% of the total variation, and the accumulative variation of the first four axes was 23.71% to 55.11% (Table 3). The first axis accounted for most of the variation of the RSCU deviation in these genes and was the main factor determining the codon usage pattern in these ciliates, the influence of the other axes being insignificant. The genes were plotted on a planar graph with the first axis as the abscissa (horizontal axis) and the second axis as the ordinate (vertical axis), respectively (Figure 5). The scattering of the genes in the graph indicated that CUB was not affected by a single factor but rather was determined by many different factors. To verify the association between CUB and nucleotide compositions, we performed a correlation analysis between nucleotide compositions and axis 1 (Appendix A). This showed a significant correlation in each species indicating that there was a correlation between CUB and nucleotide compositions. Axis 1 was significantly correlated with GC and GC3, indicating that mutation pressure was an important factor affecting CUB. In addition, CAI, ENc and axis 1 were significantly correlated with each other. Through codon correspondence analysis, we explored the codon usage patterns (Appendix A). We found that axis 1 could distinguish codons ending in G/C and A/T just as easily as axis 2 could distinguish codons ending in T/C and A/G, confirming the previous conclusion (described in the section on Nucleotide Compositions and PR2-Plot Analysis) that these ciliates preferred using codons ending in AT, especially pyrimidines.

### 3.7. Prediction of Gene Expression in 21 Ciliates Species

There is a significant positive correlation between CUB and gene expression [12,81]. The codon adaptation index (CAI) was used to predict gene expression level and codon usage bias in the 21 ciliate species. A higher CAI value means a higher gene expression level, and the CAI value range is 0 to 1 [61]. The gene expression levels of the 21 species were predicted by CAI values (Table 4). Among the 21 species, *H. grandinella* had the highest average CAI value (0.7572, SD = 0.0391), while *P. flava* had the lowest (0.5226, SD = 0.1326). The CAI values of the 21 species were all greater than 0.5, indicating that these ciliates have high gene expression levels and strong CUB. We conducted a correlation analysis between CAI and ENc values as well as between CAI and GC3 values (Figure 6 and Appendix A). With the exception of *P. carnea*, a significant negative correlation between CAI and ENc values and between CAI and GC3 values was observed, suggesting that gene expression levels may play a key role in determining the CUB in these ciliates.

### 3.8. Compositions and Gene Lengths of Amino Acids

Amino acid composition and gene length can affect CUB. Here, we conducted the correlation among Gravy and Aroma of amino acids, gene length ENc, and GC content in the genome of each of the 21 ciliate species. As shown in Appendix A, it can be seen that the Gravy of 20 species (the exception being *Tetrahymena malaccensis*) was significantly correlated with GC content (Figure 7), and the Gravy of 19 ciliate species (the exceptions being *P. caudatum* and *E. caudatum*) was significantly correlated with ENc. In the 21 ciliate species, there was a significant correlation between Aroma and GC content (Figure 8), and a very high correlation between Aroma and ENc in 16 species (the exceptions being *P. tetraurelia*, *U. marinum*, *T. malaccensis*, *S. coeruleus*, and *P. carnea*). The gene length of 17 species (the exceptions being *P. traurelia*, *P. persalinus*, *Ichthyophthirius multifiliis*, and *H. grandinella*) was significantly correlated with GC content, and the gene length of all 21 species was significantly correlated with ENc. In addition, except for *P. caudatum* and *P. persalinus*, the gene length was markedly positively correlated with ENc (Appendix A). This indicates that gene length was significantly negatively correlated with CUB. A previous study has reported that longer genes have weak CUB because selection may reduce the size of highly expressed proteins, and this effect is particularly pronounced in eukaryotes [10]. The results of the present study showed that the amino acid compositions and gene lengths could affect the CUB, but the absolute values of the correlation were low, indicating that they were only the secondary factors affecting CUB in the ciliate species investigated here.

### 3.9. Translation Selection (P2) and Choice between Pyrimidines in the 3rd Position of Codon

The P2 index, created using the principle of the distance between expected and observed codon usage, predicts the CUB using the strength of codon-anticodon binding between mRNA and tRNA [14]. The P2 index is defined as the frequency of the correct choice between pyrimidines in codons beginning with AA, AT, TA, TT, CC, CG, GC, or GG [12]. In 19 of the 21 ciliate species (the exceptions being *P. flava* and *S. stylifer*, the values of SST and WWT were higher than those of SSC and WWC (Table 5). This suggests the 3rd codon tends to end in T than C, which is consistent with the nucleotide composition as described above. Only *S. stylifer*, *P. carnea*, and *P. flava* had P2 values higher than 0.5, suggesting that translation selection played a major role in directional mutation stress in these three species, perhaps because their GC content was nearly equal to the AT content [68,82].

### 3.10. Phylogenomic Analyses

We performed phylogenetic analyses based on orthologous protein sequences of 21 ciliates to determine the relationship between ciliate systematic position and CUB (Figure 9). The phylogenetic trees constructed on BI and ML analysis had similar topologies. The result corresponded to the GC content result where in Spirotrichea the species which had higher GC content including *H. grandinella*, *P. carnea*, *P. flava* and *S. stylifer* were clustered together and other species which had lower GC content had a similar phylogenetic relationship in concatenated protein tree. Hence, our study further supported the part of Spirotrichea species that had a unique codon usage pattern and preferred using codons ending with GC.

## 4. Discussion

From prokaryotes to eukaryotes, CUB has a profound influence on genome evolution [8,83]. Contrary to Crick’s description, some ciliates do not follow the conventional protein-coding pattern of codons, but reassign termination codons to encode glutamine [54,84]. The codon is an important carrier of genetic information transmission, and the CUB in coding genes is often generated for more accurate and efficient translation. The study of CUB is therefore crucial for fully understanding the genetic and translation mechanism of ciliates [85,86].

In this study, we analyzed the CUB of 21 ciliate species representing four classes and two subphyla, and explored their molecular evolutionary mechanism so as to further understand the evolutionary relationship in different ciliate classes. Related species invariably had similar nucleotide compositions and codon usage patterns. Most species had a GC content of less than 50%, with a bias for synonymous codons ending A or T. However, the GC content of species in the class Spirotrichea was higher than that in the other classes, and the difference in GC content at the 3rd codon position was particularly significant. Measuring GC content at the 3rd codon position is a good indicator of the degree of base composition bias [78]. Based on the significant differences in GC3 content, it can be shown that there are differences in CUB among the 21 species. GC3 content and codon usage were strongly correlated among genes, suggesting that CUB may be due to a mutational bias at the DNA level rather than natural selection at the translation level. The results of nucleotide composition analysis were consistent with the PR2, RSCU and codon correspondence analyses. The CUB in most species had a bias for codons ending in AT, which contrasts with plants such as monocotyledons, which have a bias for codons ending in GC, and dicotyledons, which have a bias for codons ending in AT [87]. Due to compositional constraints, ciliates may prefer using codons ending in AT [78]. The bias in the composition of the 3rd codon base indicated that compositional constraints under mutational pressure may influence the CUB in different ciliate species. In *P. flava*, *P. carnea* and S. *stylifer*, however, nucleotide compositions, RSCU values, and putative optimal codons suggested a bias for codons ending in GC. There were only four high-frequency codons (CCT, CCA, AGA and GGA) that were common to the four ciliate classes, which suggested that CUB had large differences among the 21 species. In addition, although CUB of most genes reflects the overall AT content of the genome in *Tetrahymena thermophila*, there is a set of genes in which the optimal codon has no connection with AT content, indicating that the factors affecting ciliate CUB are complex [88].

The ENc value of ciliates was low (ENc < 40), which can indicate that many ciliates may need high gene expression to adapt to environmental stress. Furthermore, ENc-plots showed that there was a significant correlation between ENc and GC3s, which indicated that mutation pressure existed in the CUB. However, some genes in ENc-plots were far from the curve, indicating that mutation pressure did not play a major role in CUB. The significant negative correlation between ENc and CAI, and the significant negative correlation between CAI and GC3 indicated that genes with high expression had stronger CUB, and that gene expression was one of the most important factors for CUB. Surprisingly, however, in *Tetrahymena* genes with high expression had higher GC content and tended to have codons ending in GC. It has previously been speculated that codons ending in GC have higher translation efficiency and accuracy [48,49].

Correspondence analysis showed that nucleotide composition, which plays an important role in CUB, is significantly correlated with axis 1. Furthermore, correlation analysis indicated that multiple factors such as gene length and Gravy and Aroma of amino acids together influenced CUB in the 21 ciliate species. PR2 analysis showed that mutation pressure and natural selection bias were both involved in CUB. The neutral theory of molecular evolution suggests that silent mutation sites in codons represent neutral evolution [89]. In this study, GC12 and GC3 showed a significant correlation, indicating that mutation pressure plays an important role in CUB in ciliates. However, the linear regression slope of the neutral plot was less than 50%, suggesting that natural selection bias may also play a major role.

The high GC content in the genome of the class Spirotrichea may be due to environmental stress, resulting from stable DNA. Biased gene transformation (BGC) or mutation pressure that changed AT into GC may be the reason for the differences in nucleotide compositions [50]. The BGC is a GC-biased repair process occurring in the recombinant genome, which is the main driving force of genome evolution [90]. CUB in ciliates may be an adaptive mechanism to facilitate adaptation to environmental conditions. Therefore, ciliates from different environments may differ in their CUB.

## 5. Conclusions

Though different species in ciliate have variant genome size and GC content, we conclude that most of the ciliates investigated prefer using the codons of AT-ending and the CUB of ciliates is affected by gene mutation and natural selection together.

## Figures and Tables

**Figure 1 microorganisms-11-01833-f001:**
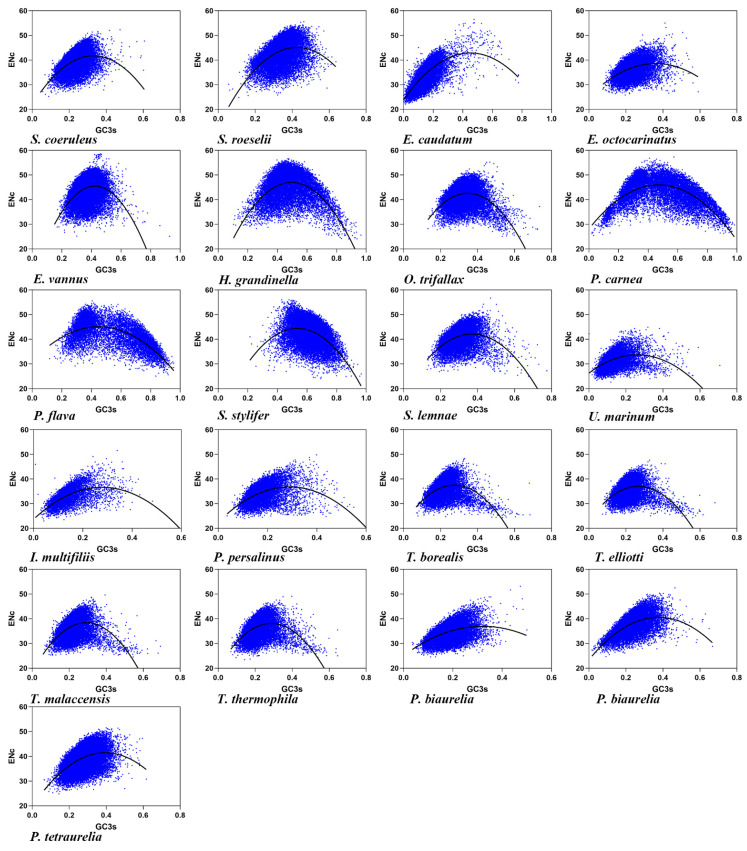
Relationship between the ENc and GC3 content at the third codon position of the 21 ciliate species. Blue dots represent the genes.

**Figure 2 microorganisms-11-01833-f002:**
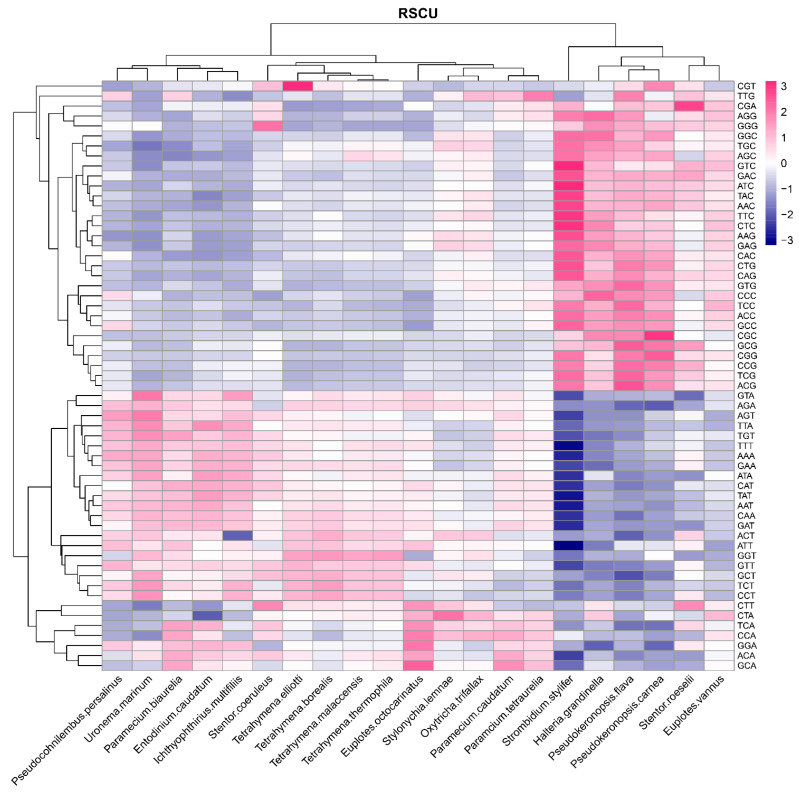
The average RSCU values of the codons in the 21 ciliate species. A gradient dark blue to dark red indicates the RSCU value increases from low to high.

**Figure 3 microorganisms-11-01833-f003:**
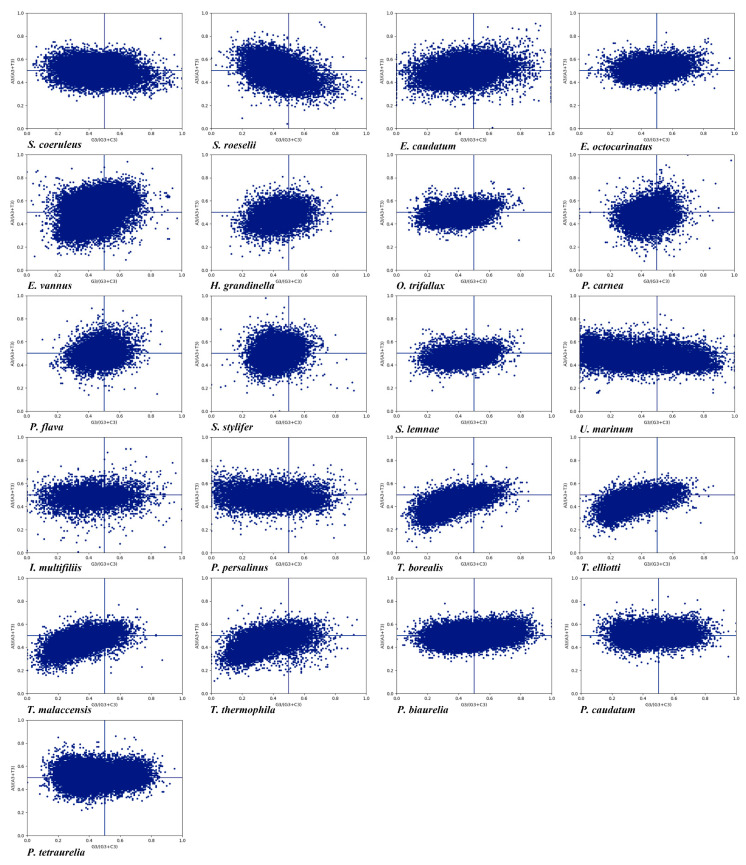
PR2-bias plot of A3/(A3 + T3) against G3/(G3 + C3) in 2-fold, 4-fold, and 6-fold degenerate amino acids in the 21 ciliate species. Blue dots represent the genes.

**Figure 4 microorganisms-11-01833-f004:**
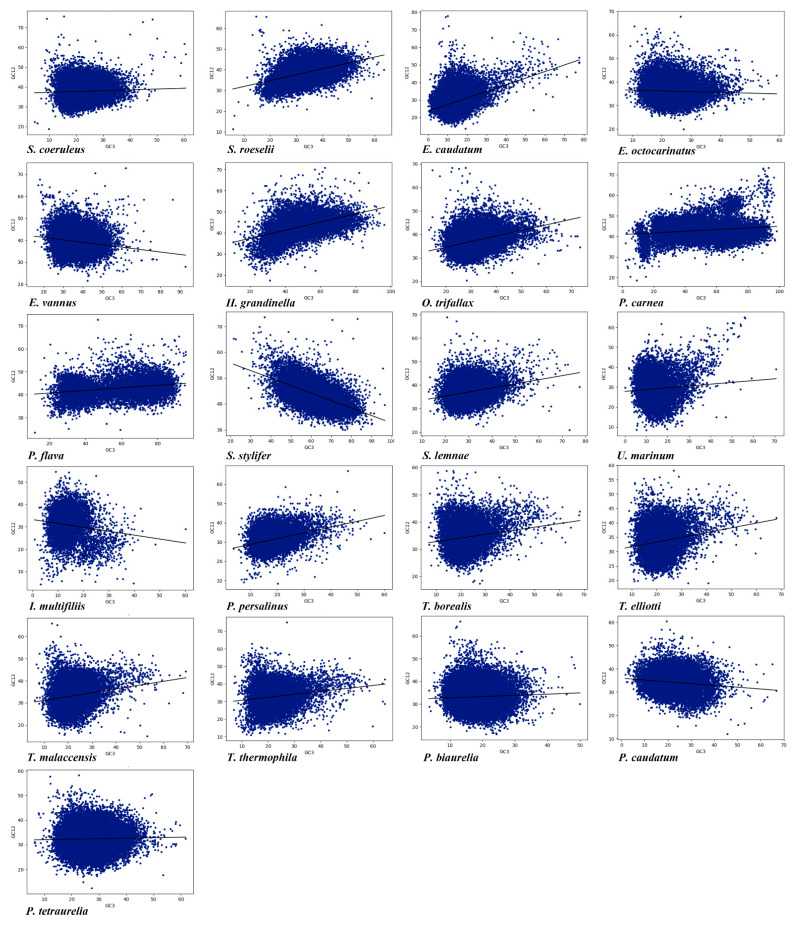
Neutrality plots, showing the relationship between GC3 and GC12 in the 21 ciliate species. Blue dots represent the genes.

**Figure 5 microorganisms-11-01833-f005:**
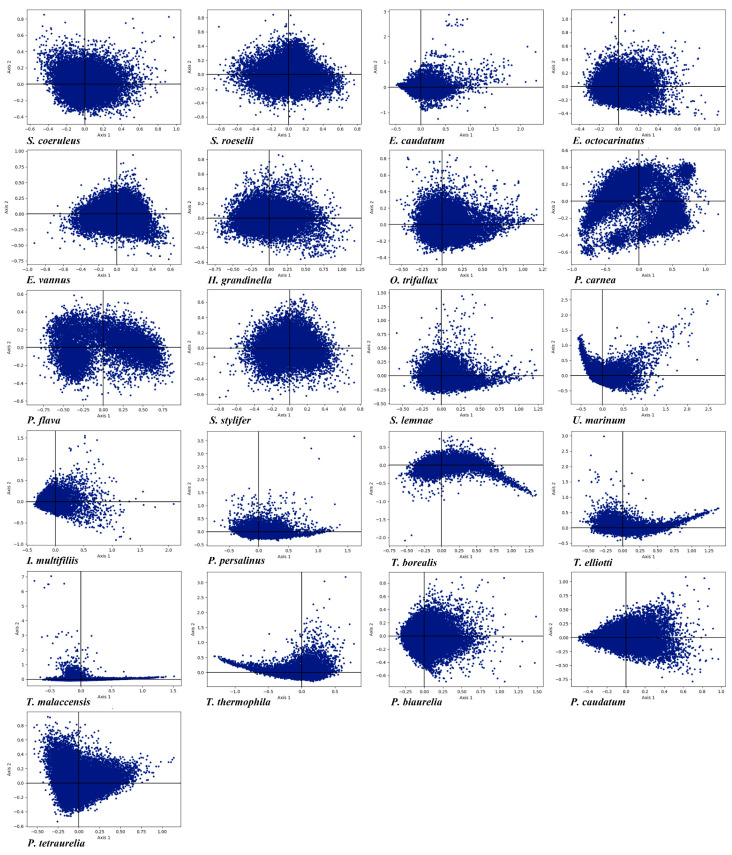
RSCU correspondence analysis plot of each gene in the 21 ciliate species. Axis 1 and axis 2 represent the largest contributors to the RSCU values of genes.

**Figure 6 microorganisms-11-01833-f006:**
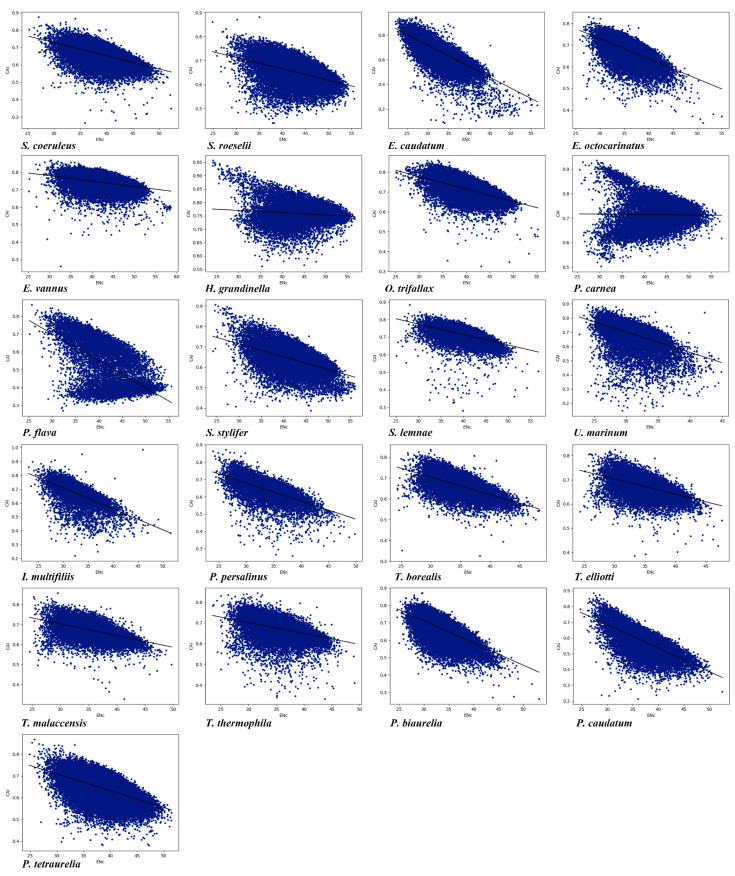
Relationship between the CAI and the ENc of the 21 ciliate species. Blue dots represent the genes.

**Figure 7 microorganisms-11-01833-f007:**
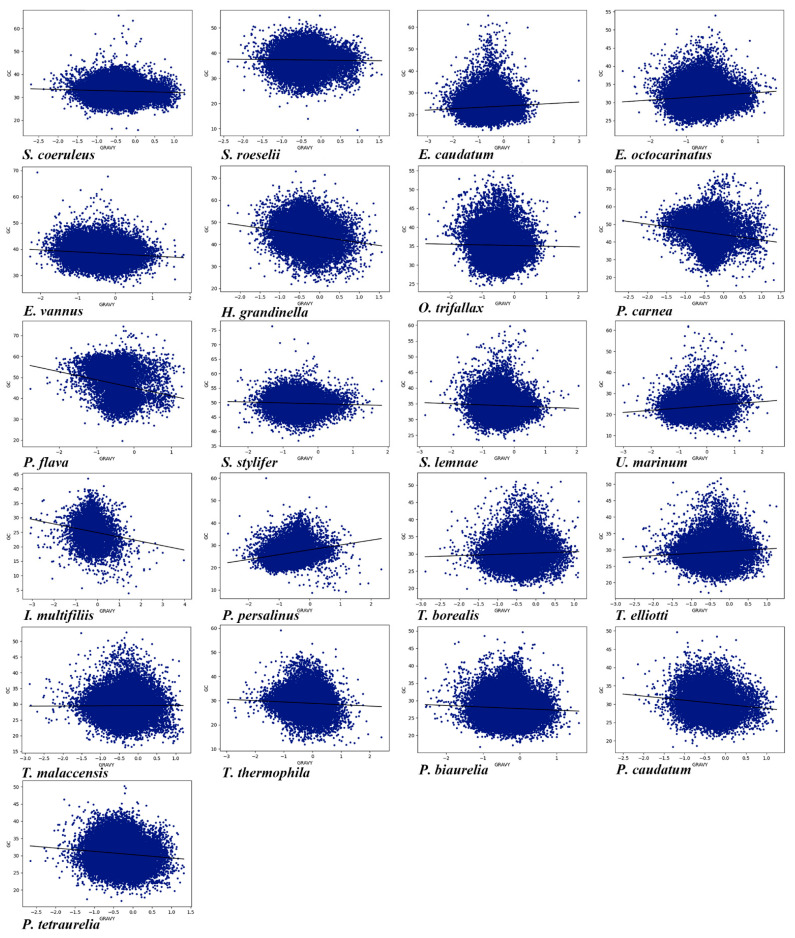
Relationship between Gravy and the overall content of GC in the 21 ciliate species. Blue dots represent the genes.

**Figure 8 microorganisms-11-01833-f008:**
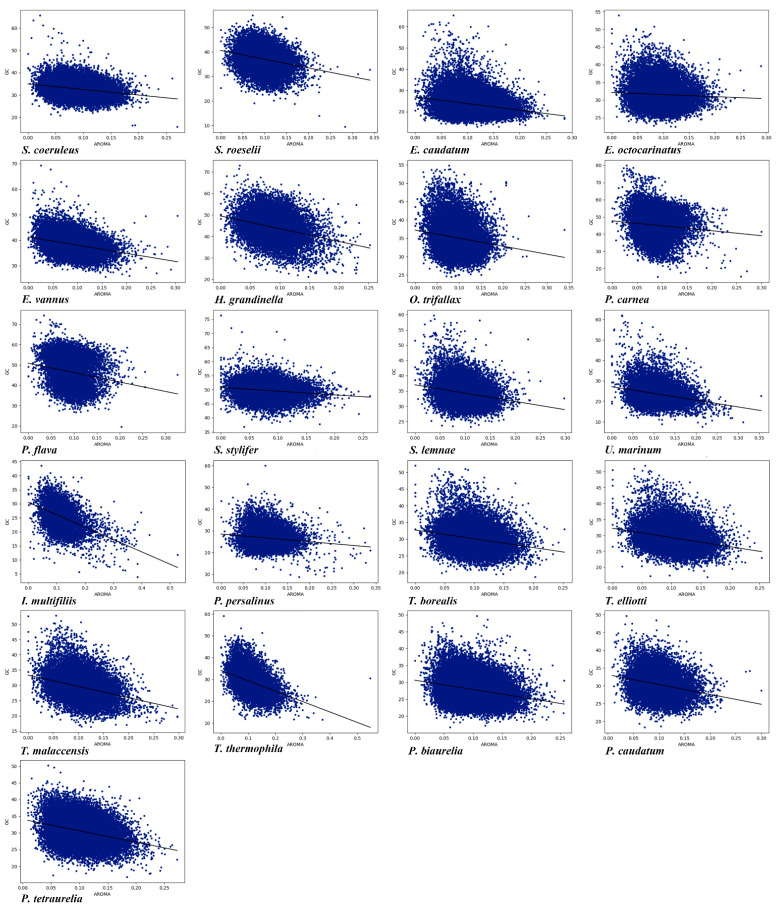
Relationship between Aroma and the overall content of GC in the 21 ciliate species. Blue dots represent the genes.

**Figure 9 microorganisms-11-01833-f009:**
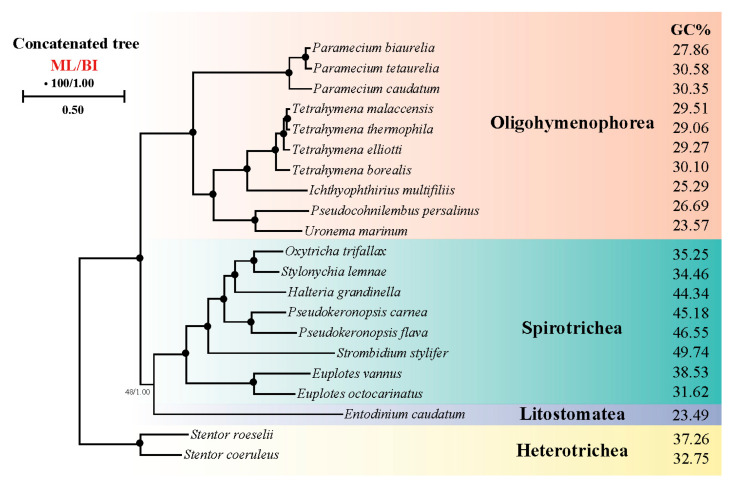
Phylogenetic tree generated from maximum likelihood (ML) and Bayesian inference (BI) based on a concatenation of orthologous protein sequences. Numbers near nodes represent bootstrap values of ML and posterior probabilities of BI. Fully supported (100/1.00) nodes are marked with solid circles. The scale bar corresponds to one substitution per two nucleotide sites.

**Table 1 microorganisms-11-01833-t001:** Nucleotide composition of the 21 ciliate species.

Species	A3%	T3%	C3%	G3%	GC%	GC1%	GC2%	GC12%	GC3(s)%
*Stentor roeselii*	31.91	33.87	19.34	14.88	37.26	44.26	33.29	38.77	34.22
*Stentor coeruleus*	38.05	39.29	12.4	10.25	32.75	42.79	32.80	37.79	22.66
*Euplotes vannus*	33.13	30.16	20.20	16.51	38.53	41.45	37.48	39.45	36.71
*Euplotes octocarinatus*	39.51	37.90	12.14	10.45	31.62	41.56	30.70	36.13	22.59
*Strombidium stylifer*	19.40	18.61	35.78	26.20	49.74	52.32	34.91	43.61	61.98
*Halteria grandinella*	25.13	27.26	27.21	20.40	44.34	48.67	36.72	42.70	47.61
*Stylonychia lemnae*	33.23	37.19	16.69	12.89	34.46	42.10	31.70	36.90	29.58
*Oxytricha trifallax*	33.15	35.51	18.31	12.95	35.25	42.32	32.15	37.21	31.30
*Pseudokeronopsis flava*	22.32	23.32	28.54	25.82	46.55	50.97	34.32	42.65	54.36
*Pseudokeronopsis carnea*	23.81	26.40	27.60	22.20	45.18	51.67	34.07	42.87	49.79
*Entodinium caudatum*	42.95	43.72	7.61	5.78	23.49	32.93	24.20	28.56	13.33
*Tetrahymena borealis*	34.45	43.87	13.60	8.08	30.10	37.88	30.74	34.31	21.68
*Tetrahymena elliotti*	35.75	43.37	13.24	7.65	29.27	37.30	29.63	33.46	20.88
*Tetrahymena malaccensis*	35.54	42.38	14.05	8.03	29.51	36.89	29.56	33.22	22.09
*Tetrahymena thermophila*	36.89	36.06	14.57	12.49	29.06	36.10	29.32	32.71	21.76
*Ichthyophthirius multifiliis*	41.19	44.54	8.22	6.05	25.29	34.70	26.84	30.77	14.27
*Pseudocohnilembus persalinus*	39.94	42.21	11.04	6.81	26.69	35.65	26.56	31.10	17.85
*Uronema marinum*	41.76	45.43	7.87	4.95	23.57	32.83	25.09	28.96	12.80
*Paramecium biaurelia*	40.84	41.88	8.96	8.31	27.86	37.04	29.27	33.15	17.00
*Paramecium caudatum*	39.35	38.08	11.91	10.67	30.35	38.38	30.09	34.24	22.57
*Paramcium tetraurelia*	36.89	36.06	14.57	12.49	30.58	35.14	29.56	32.35	27.06

**Table 2 microorganisms-11-01833-t002:** The effective number of codons of the 21 ciliate species.

Species	Range	Mean	SD
*Stentor roeselii*	24.8233~55.6122	43.2310	4.7995
*Stentor coeruleus*	25.2281~52.2843	39.4773	3.3917
*Euplotes vannus*	25.1212~58.6122	44.2310	3.9264
*Euplotes octocarinatus*	25.8303~55.0067	36.4383	2.8887
*Strombidium stylifer*	24.1399~56.0129	42.6091	4.7208
*Halteria grandinella*	24.0406~56.6128	44.8589	5.4764
*Stylonychia lemnae*	25.0396~56.7110	40.5681	3.4984
*Oxytricha trifallax*	25.0583~55.2462	41.5530	3.5766
*Pseudokeronopsis flava*	25.1127~55.5793	41.8680	5.2695
*Pseudokeronopsis carnea*	25.4563~57.3212	42.6069	5.2678
*Entodinium caudatum*	22.1304~56.4895	33.2936	3.5350
*Tetrahymena borealis*	54.3635~48.3126	36.2745	3.2871
*Tetrahymena elliotti*	24.2852~47.6539	35.8999	3.0246
*Tetrahymena malaccensis*	24.5190~49.6401	36.6933	3.4687
*Tetrahymena thermophila*	23.4909~49.0903	36.5193	3.3741
*Ichthyophthirius multifiliis*	22.9875~51.5511	33.0615	3.3160
*Pseudocohnilembus persalinus*	23.8975~49.8253	34.2206	3.4134
*Uronema marinum*	24.8432~51.4703	38.8706	3.8384
*Paramecium biaurelia*	24.5789~53.1388	34.2286	2.7692
*Paramecium caudatum*	24.7160~52.5174	34.2473	3.8172
*Paramcium tetraurelia*	22.6886~44.8700	31.4826	2.5502

**Table 3 microorganisms-11-01833-t003:** Variation in correspondence analysis of the 21 ciliate species.

Species	Axis1	Axis1-4
*Stentor roeselii*	11.40%	31.06%
*Stentor coeruleus*	7.30%	29.02%
*Euplotes vannus*	8.46%	25.84%
*Euplotes octocarinatus*	5.82%	31.80%
*Strombidium stylifer*	8.71%	34.37%
*Halteria grandinella*	18.48%	36.86%
*Stylonychia lemnae*	9.72%	24.75%
*Oxytricha trifallax*	13.49%	23.71%
*Pseudokeronopsis flava*	32.40%	49.76%
*Pseudokeronopsis carnea*	37.59%	48.13%
*Entodinium caudatum*	9.07%	36.80%
*Tetrahymena borealis*	16.41%	30.54%
*Tetrahymena elliotti*	13.46%	31.80%
*Tetrahymena malaccensis*	11.74%	34.70%
*Tetrahymena thermophila*	10.54%	36.96%
*Ichthyophthirius multifiliis*	9.23%	48.71%
*Pseudocohnilembus persalinus*	16.36%	55.11%
*Uronema marinum*	10.86%	51.81%
*Paramecium biaurelia*	7.01%	34.35%
*Paramecium caudatum*	11.00%	34.63%
*Paramcium tetraurelia*	8.08%	36.31%

**Table 4 microorganisms-11-01833-t004:** The codon adaptation index value of the 21 ciliate species.

Species	Range	Mean	SD
*Stentor roeselii*	0.4404~0.8811	0.6505	0.0516
*Stentor coeruleus*	0.2648~0.8749	0.6570	0.0502
*Euplotes vannus*	0.2612~0.8672	0.736	0.0344
*Euplotes octocarinatus*	0.334~0.8298	0.6700	0.0469
*Strombidium stylifer*	0.3869~0.9063	0.6354	0.0583
*Halteria grandinella*	0.5605~0.9565	0.7572	0.0391
*Stylonychia lemnae*	0.2803~0.8844	0.7117	0.0401
*Oxytricha trifallax*	0.3252~0.8593	0.7054	0.0402
*Pseudokeronopsis flava*	0.2695~0.8644	0.5226	0.1326
*Pseudokeronopsis carnea*	0.5025~0.9352	0.7142	0.0610
*Entodinium caudatum*	0.084~0.9423	0.6605	0.0832
*Tetrahymena borealis*	0.3253~0.8352	0.6524	0.0447
*Tetrahymena elliotti*	0.3861~0.8234	0.6663	0.0375
*Tetrahymena malaccensis*	0.3268~0.8574	0.6643	0.0381
*Tetrahymena thermophila*	0.3325~0.8438	0.6673	0.0433
*Ichthyophthirius multifiliis*	0.2185~0.9845	0.6617	0.0833
*Pseudocohnilembus persalinus*	0.2584~0.8712	0.6359	0.0593
*Uronema marinum*	0.1479~0.8936	0.6806	0.0812
*Paramecium biaurelia*	0.2625~0.8712	0.6574	0.0595
*Paramecium caudatum*	0.2127~0.8852	0.5774	0.0848
*Paramcium tetraurelia*	0.3803~0.8669	0.6405	0.0563

**Table 5 microorganisms-11-01833-t005:** The P2 indices of the 21 ciliate species.

Species	WWC	WWT	WWY	SST	SSC	SSY	P2
*Stentor roeselii*	37.69	55.54	93.23	20.44	7.57	28.01	0.4906
*Stentor coeruleus*	22.70	63.02	85.72	24.25	4.68	28.93	0.4190
*Euplotes vannus*	34.21	50.08	84.29	17.56	10.57	28.13	0.4559
*Euplotes octocarinatus*	23.14	65.13	88.27	15.17	3.01	18.18	0.3680
*Strombidium stylifer*	45.48	15.65	61.13	13.36	16.06	29.42	0.6492
*Halteria grandinella*	36.46	36.62	73.08	17.05	19.90	36.95	0.4794
*Stylonychia lemnae*	41.92	84.27	126.19	25.99	9.94	35.93	0.4222
*Oxytricha trifallax*	47.00	81.83	128.83	27.04	11.46	38.50	0.4454
*Pseudokeronopsis flava*	34.85	37.49	72.34	14.95	17.14	32.09	0.5006
*Pseudokeronopsis carnea*	46.97	54.43	101.40	28.16	26.77	54.93	0.5101
*Entodinium caudatum*	20.02	134.57	154.59	17.74	4.13	21.87	0.2231
*Tetrahymena borealis*	40.34	118.10	158.44	36.07	7.58	43.65	0.3850
*Tetrahymena elliotti*	36.80	113.12	149.92	31.12	6.27	37.39	0.3688
*Tetrahymena malaccensis*	36.84	105.11	141.95	28.84	6.53	35.37	0.3712
*Tetrahymena thermophila*	37.91	115.27	153.18	29.81	6.46	36.27	0.3623
*Ichthyophthirius multifiliis*	15.66	88.17	103.83	19.21	3.73	22.94	0.2805
*Pseudocohnilembus persalinus*	23.97	106.22	130.19	18.74	7.57	26.31	0.2761
*Uronema marinum*	22.8	137.77	160.57	25.51	4.25	29.76	0.2679
*Paramecium biaurelia*	18.54	86.69	105.23	19.38	3.6	22.98	0.3057
*Paramecium caudatum*	22.94	77.54	100.48	18.01	5.26	23.27	0.3348
*Paramcium tetraurelia*	28.64	77.53	106.17	13.69	6.02	19.71	0.3346

## Data Availability

The datasets presented in this study can be found in National Center for Biotechnology Information DataBase (NCBIdb) with accession number shown Table 1.

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
