# Peer review of "Codon Usage Bias Analysis in Macronuclear Genomes of Ciliated Protozoa"

_microorganisms, 2023, doi:10.3390/microorganisms11071833_

Round 1

Reviewer 1 Report

The reviewed paper dealt with the analysis of the codon usage bias (CUB) of the macronuclear genome of 21 ciliate species from four classes, namely Heterotrichea, Spirotrichea, Litostomatea, and Oligohymenophorea. The study was carried out at a high methodological level, important results were obtained, indicating that the codons of AT-ending and the CUB of ciliates is simultaneously affected both by gene mutations and natural selection. These results complementing the understanding of evolutionary processes in ciliates as well as help to improve the efficiency of gene editing in model ciliates.

Reviewer 2 Report

This manuscript, entitled Codon Usage Bias Analysis in Macronuclear Genomes of Ciliated Protozoa by Y. Fu et al., describes a statistical comparison of the genomic structure, especially in the codon usage bias, of 21 ciliate species. The reviewer acknowledges the importance is the comparative analysis of the genomes of 21 phylogenetically diverse ciliate species. However, the presentation of the results is insufficient and needs to be improved. For example, in Figures 1-4 and 6-10, the species name should be added to the top of each graph, and Table 1, which showed a list of genome data used in the analysis, should be included in the supplement. The points indicated by Figures 1 and 2 are fully explained in the text and Table 2, and these two figures should be included in the Supplement. Figure 4, it is sufficient to show a representative example in the text. Figure 8 should also be included in the Supplement. Besides, the reviewer felt that the manuscript presented the results of various analyses in a rambling manner. The reviewer also pointed out that the paper needs to discuss the relationships of the ciliate phylogeny and their findings, because the authors applying various ciliates for the statistical comparison examined here. Phylogenetic relationships among ciliates have been studied repeatedly, and established phylogenetic trees equipping good resolution have been constructed (e.g., Gao. F. et al., 2016, DOI: 10.1038/srep24874). The reviewer considers it is essential to compare the authors' statistical results with the phylogenetic relationships of ciliates. For the above points, the reviewer cannot recommend publishing this paper in the journal Microorganisms with the current submitted form. Major revisions are needed.

Reviewer 3 Report

1. The authors used the complete genome sequences of 21 ciliate species to analyse codon usage. In addition to these 21 ciliate genomes, there are many ciliate whole genome sequences available in the NCBI database. I would like to know why the authors chose these particular 21 ciliates for the study.

2. The authors used different methods to study the codon preferences of genes from the 21 ciliate species, which belong to four different classes. I advise the authors to investigate possible differences in codon usage between the four classes.

3. One question that arises concerns the codon adaptation index analysis, which requires the use of a reference codon usage table. The authors should explain in the method section how to construct the reference codon usage table or provide a download link for the reference codon usage table.

4. In the Results section "3.1. Subsection Nucleotide Compositions", the authors claimed that "There was a significant difference in GC content..." and "These data indicated that there were significant differences...". In fact, the authors did not perform statistical analysis. Therefore, the use of the term "significant difference" is inappropriate. The authors should revise such inappropriate statements throughout the manuscript.

5. In Table S3, the preferred codon, overrepresented codons (RSCU > 1.6), underrepresented codons (RSCU < 0.6) and optimal codons for each species should be highlighted using typographical emphasis (e.g. bold, italic and underlined formatting).

Round 2

Reviewer 2 Report

Regarding the previous point 2, the response to the reviewers was as follows: "We have revised the article to make the results clearer. •••ã€€The methods of analysis were ranked by their importance. For the details, see the text, please.

After carefully reviewing the revised article, the reviewer has concluded that apart from the phylogenetic analysis result and some figures transferred to the Supplement, there were no noticeable revisions to the main text.  If the authors do not agree with the reviewer's comments, they should clearly state and refute them.

The reviewer sincerely appreciates the authors acknowledged the previous comment about the phylogenetic relationships and included the analysis. However, the reviewer cannot accept that the authors submitted an unchanged manuscript as a revised version and claimed it to be "clearer." 

As a result, the reviewer maintains the opinion that the manuscript needs to be revised.
